# Cas9-derived peptides presented by MHC Class II that elicit proliferation of CD4+ T-cells

Vijaya L. Simhadri[1], Louis Hopkins[1], Joseph R. McGill [1], Brian R. Duke[2], Swati Mukherjee[2], Kate Zhang[2] &
Zuben E. Sauna [1✉]

CRISPR–Cas9 mediated genome editing offers unprecedented opportunities for treating human diseases. There are several reports that demonstrate pre-existing immune responses to Cas9 which may have implications for clinical development of CRISPR-Cas9 mediated gene therapy. Here we use 209 overlapping peptides that span the entire sequence of Staphylococcus aureus Cas9 (SaCas9) and human peripheral blood mononuclear cells (PBMCs) from a cohort of donors with a distribution of Major Histocompatibility Complex (MHC) alleles comparable to that in the North American (NA) population to identify the immunodominant regions of the SaCas9 protein. We also use an MHC Associated Peptide Proteomics (MAPPs) assay to identify SaCas9 peptides presented by MHC Class II (MHC-II) proteins on dendritic cells. Using these two data sets we identify 22 SaCas9 peptides that are both presented by MHC-II proteins and stimulate CD4+ T-cells.

[1] Hemostasis Branch, Division of Plasma Protein Therapeutics, Office of Tissues and Advanced Therapies, Center for Biologics Evaluation and Research, Food and Drug Administration, Silver Spring, MD, USA. [2] Editas Medicine, Cambridge, MA, USA. ✉email: zuben.sauna@fda.hhs.gov

Gene-editing technologies based on Cas proteins have recently entered clinical development[1,2]. Since the early days of the development of the CRISPR Cas9 gene-editing technology, there have been speculations that immune responses to Cas9 (which is of microbial origin) could have consequences for clinical applications[3,4]. Since 2018, studies have presented experimental evidence for T- and B-cell responses to Cas9[1,5–8] derived from Staphylococcus aureus (SaCas9) and Streptococcus pyogenes (SpCas9). In these studies, anti-Cas9 antibodies were detected in the serum samples of a fraction of healthy donors[5,6]. In addition, both the CD4+ and CD8+ T-cells from these donors respond to Cas9 proteins[5,7]. Consistent with these findings, in silico assessment of the Cas9 protein indicates that several peptides derived from SpCas9 bind to MHC proteins with high affinity[9]. Although these and other studies present evidence of memory T-cells to Cas9 in the human population[3–5,7,9] due to exposure to *Staphylococcus*[5] the identification of biologically/clinically meaningful T-cell epitopes has not been accomplished, i.e., Cas9-derived peptides that are generated by the proteolytic machinery of antigen-presenting cells (APCs), presented by MHC molecules and elicit T-cell proliferation.

Immune responses to proteins which are used in therapeutic applications can affect the safety and/or efficacy of these products. Consequently, immunogenicity risk assessments are required during drug development and licensure[10]. Per FDA guidance, the Cas proteins are likely to be in the high-risk category because they were originally isolated from human pathogens[11]. Immunogenicity risk assessments strive to answer two fundamental questions: (i) What is the probability of eliciting an immune response to the protein that is used therapeutically? (ii) What is the clinical consequence of such an immune response? The studies described in the previous paragraph can help to answer the first question. Very little is known about the clinical consequences of an anti-Cas9 immune response. A recent mouse animal study offers a glimpse into the in vivo consequences of pre-existing immunity to the Cas proteins. The liver genome editing with AAV packaging CRISPR-Cas9 was determined to occur efficiently even in the presence of pre-existing SaCas9 immunity. Interestingly, however, genome editing was accompanied by an increase in CD8+ T-cells in the liver and a cytotoxic T-cell response. These immune responses resulted in hepatocyte apoptosis, loss of recombinant AAV genomes, and complete elimination of genome-edited cells[12].

T-cell responses to Cas9 can be critical in evaluating the risks associated with different Cas proteins that will be used therapeutically. Moreover, if fit-for-purpose clinical assays are developed to measure T-cell responses to Cas proteins, they can be used to profile potential patients who are at high risk of severe adverse events. The identification of T-cell epitopes is key to a comprehensive understanding of the immunogenicity of Cas proteins. A widely used technique to identify epitopes relies on measuring T-cell responses following incubation of peripheral blood mononuclear cells (PBMCs) with overlapping peptides derived from the protein of interest[10]. While the use of peptides (or peptide-pools) allows the identification of T-cell epitopes and potential hot-spots on the protein sequence, there is one critical drawback to this approach when it is used in isolation. Peptides are presented to T-cells following processing (i.e., proteolytic cleavage). As the entire repertoire of overlapping peptides will not be generated in vivo, only a sub-set of putative T-cell epitopes will be presented by antigen-presenting cells, such as dendritic cells (DCs)[13,14]. The mass spectrometry (MS)-based MHC associated peptide proteomics (MAPPs) assay is an analytical tool that identifies the peptides from the protein that are naturally processed and presented on the MHC molecules expressed by antigen-presenting cells[15,16]. The MAPPs assay thus provides information about both the protein-processing and peptide-presentation.

Here we present a strategy to identify SaCas9-derived peptides that are processed by DCs, presented by MHC-II molecules, and stimulate CD4+ T-cells. Similar strategies can be designed where SaCas9 proteins are expressed in antigen-presenting cells, and peptides expressed on MHC Class I (MHC I) proteins that stimulate CD8+ T-cells are identified but are not part of this study.

## Results

**Experimental workflow**. This study was carried out in three parts. Part 1: a set of 209 overlapping peptides spanning the primary sequence of SaCas9 (Supplementary Fig. 1) were used to identify peptides that elicited CD4+ T-cell proliferation in PBMCs sourced from 21 healthy donors. Part 2: the purified, endotoxin-free, full-length SaCas9 protein was used to identify SaCas9-derived peptides presented by MHC Class II (MHC-II) proteins on DCs using the MAPPs assay. Part 3: the two datasets were merged to identify SaCas9-derived peptides that are processed and presented by antigen-presenting cells (the DCs) and induce proliferation of CD4+ T-cells.

**SaCas9 peptides that elicit CD4+ T-cell proliferation**. For Part 1, we carefully collated 21 donors as a source of PBMCs for the ex vivo assays. These samples were selected from a larger pool of 50 MHC-typed donors using SampPick[17] to closely match the frequency distribution of MHC variants the North American (NA) Population (Fig. 1a).

The peptides were organized into 21 pools of 10 peptides each (Pool 21 has nine peptides) (Supplementary Fig. 1). T-cell responses to each peptide pool was monitored individually in PBMCs from each of the 21 donors. Donor demographics and HLA type are provided in Supplementary Table 1. The workflow and gating strategy for the flow cytometry-based experiment is depicted in Supplementary Fig. 2. Following ex vivo stimulation of PBMCs with peptide-pools, SaCas9 proteins or controls, intracellular staining of CD4+ T-cells was used to measure cytokines associated with T-cell activation, viz., interferon-γ (IFN-γ), tumor necrosis factor-α (TNF-α), and interleukin 2 (IL-2). The functions of these markers are provided in Supplementary Table 2. The number of donors whose CD4+ T-cells test positive for each marker following stimulation are depicted in Fig. 1b (see Methods for details of the statistical methodology). Using these data, we determined the promiscuity of each peptide pool, i.e., the percent of donors whose T-cells were stimulated (Fig. 1c). Here, we show a worst-case scenario wherein a donor is considered positive if any one of the three cytokines measured elicited a positive response. Based on our criterion of what constitutes a T-cell response, any individual peptide-pool elicited a response in 9.5–52% of the cohort and the Sa- and SpCas9 proteins elicited a response in 14–53% of the cohort.

**Individual donors respond differently to SaCas9 peptides**. To determine the plausible immunogenicity profile for an individual donor we determined the number of peptide pools that elicit a T-cell response in each donor. We used the rule described in the previous section to determine which peptide-pool donor combinations resulted in a positive response. There are sharp differences in the response of individual donors (Fig. 2a). CD4+ T-cells from two donors (Donors 3 and 9) responded to 14 of the 21 peptide pools (66.6%). Donors 5 and 12 on the other hand responded to only one peptide pool each (~5% of peptide pools). These results indicate that T-cell responses are highly variable across individuals.

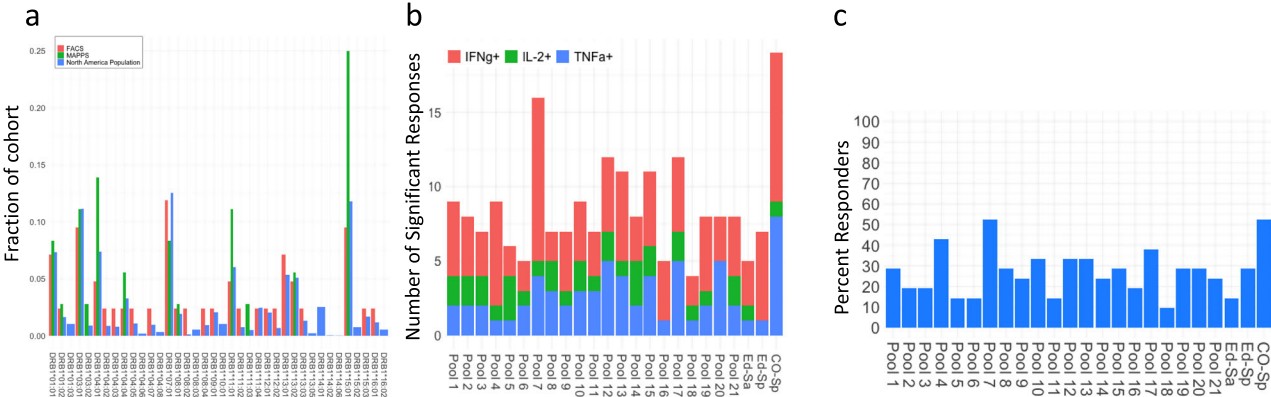

**Fig. 1 SaCas9 specific peptide pools and their T-cells responses in North American population. a**, A comparison of allele frequencies in the North American population (blue), the cohort of donors selected by the SampPick algorithm for the flow cytometry assay (red), and the donors included in the MAPPs assay (green). **b**, Responses of CD4$^+$ T-cells obtained from 21 donors to peptide pools from SaCas9 and three full-length Cas9 proteins are shown. The three Cas9 proteins are SaCas9 from Editas (Ed-Sa), SpCas9 from Editas (Ed-Sp) and control SpCas9 from a commercial vendor (CO-Sp). Significant responses were identified using a one-sided Fisher's exact test comparing the cell counts for IFN-γ (red), TNF-α (blue), and IL-2 (green) in CD4$^+$ T-cells as compared to unstimulated samples. *P* values were adjusted according to the Bonferonni-Holm method. **c**, Percent of donors responsive to each of the peptide pools and Sa- and SpCas9 proteins (see **b**). Donors were considered responders if at least one of the three cytokines (IFN-γ, TNF-α, or IL-2) were significantly higher than cell counts for the unstimulated samples.

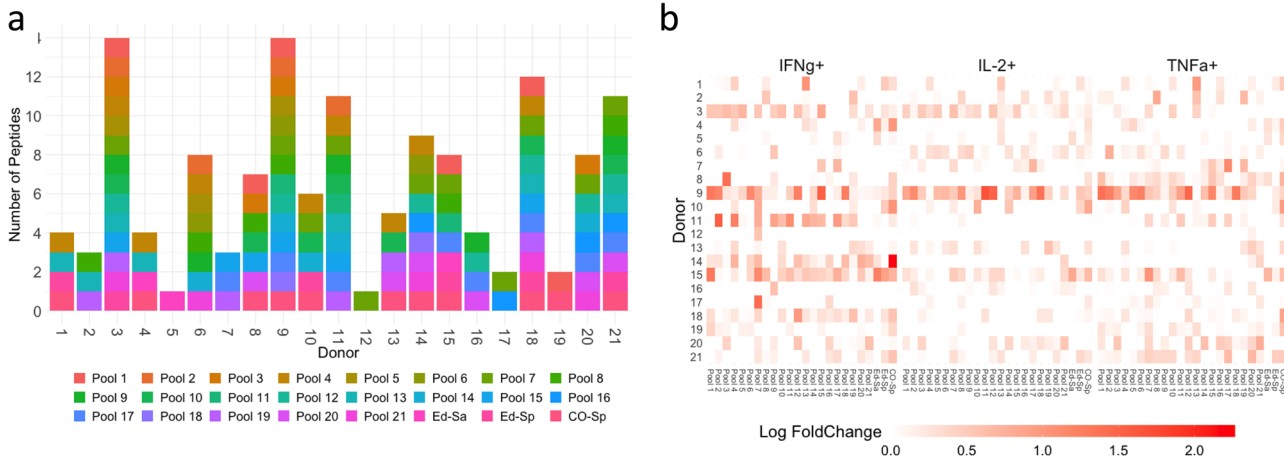

**Fig. 2 Differences in T -cell activation across the donors. a** The number of peptide-pools that induced CD4$^+$ T-cell activation in PBMCs from each donor is shown. The peptide pools that elicit responses are color coded by peptide-pool number. **b** Heatmap showing the Log fold change for each biomarker compared to unstimulated controls (i.e., cells were not exposed to any Cas9 protein or peptide) in CD4$^+$ cells.

The analyses described in the previous paragraph are based on evaluating whether an individual peptide-pool-donor combination activated CD4$^+$ T-cells. Among activated T-cells, we also quantified the T-cell response vis-à-vis the individual markers as compared to an unstimulated sample (see Methods). The intensity of the T-cell response varies greatly between donors and T-cell activation markers (Fig. 2b).

**MHC-II variants associated with Cas9 activation of T-cells**. A further important consideration while identifying putative T-cell epitopes in the Cas9 protein is that stimulation of CD4$^+$ T-cells involves the presentation of foreign peptides by MHC-II proteins. We used HLA-typed donors as a source of PBMCs (Fig. 1a, Supplementary Table 1) and this allows us to evaluate the MHC-II variants associated with the peptide-pool driven activation of CD4$^+$ T-cells. Here, we considered only the two DRB1 alleles present per donor. We consider this a reasonable assumption as previous studies[18] have demonstrated that immune responses to

therapeutic proteins are predominantly mediated by the DRB1 alleles. We used the criterion for identifying T-cell activation described above (i.e., response positive for one of the three activation markers). The MHC-II-peptide-pool pair was identified as driving T-cell activation under two scenarios: (i) a less stringent measure wherein cells from any donor with that MHC-II allele showed T-cell activation (Fig. 3a, upper panel) or (ii) a more rigid criterion wherein cells from all donors with that MHC-II allele showed T-cell activation (Fig. 3b, upper panel). The results show that CD4$^+$ T-cell activation is strongly dependent on the MHC-II variant expressed on the APCs. The binary maps (Fig. 3a, b, upper panels) show which peptide-pool-MHC combinations are associated with T-cell activation. The underlying data in these binary maps can be used to gain useful information about the potential immunogenicity of the Cas9 protein in the population. To do this we used a promiscuity score[15,19], that computes the number of MHC variants each peptide pool positively interacted with and assigns weights based on the frequency of each MHC variant in the NA population (Fig. 3a, b, lower panels). There are,

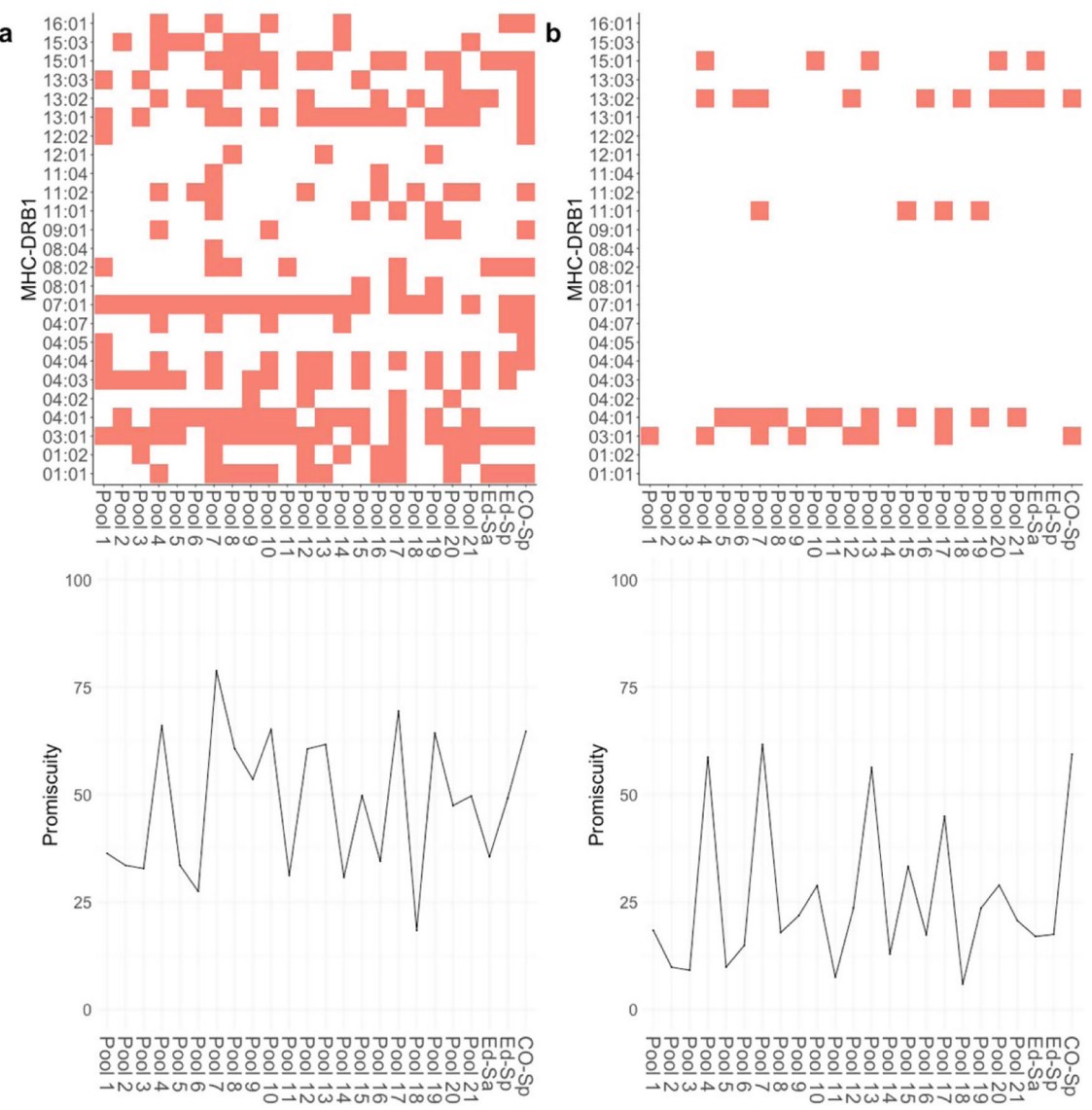

**Fig. 3 Identification MHC I and MHC-II-mediated T -cell immune responses after peptide stimulation. a** Upper panels, MHC-II alleles that are associated with CD4[+] activation. An MHC was deemed to be associated with a T-cell response if cells from at least one donor with that MHC allele was activated by a peptide pool. **b** Upper panels, MHC-II alleles that are associated with CD4[+] activation selected based on the rule that an MHC allele is associated with T-cell activation only if all donors with that allele have a positive response for a peptide pool. Binary plots show which MHC alleles are considered to be responders to each pool. **a**, **b** Lower panels. Promiscuity plots for data presented in the upper panels. The promiscuity plot shows the number of MHC variants each peptide pool is associated with, weighted for the frequency with which each allele occurs in the NA population.

of course, numerous caveats and limitations to these estimates but this measure can provide an approximation of the proportion of the population that could show T-cell immune responses to Cas9.

**SaCas9 peptides processed by DCs and presented by MHC-II.** In Part 2 of this study, we carried out an MAPPs assay to identify SaCas9 peptides processed and presented by MHC-II proteins on DCs. The assay was carried out using DCs from 18 healthy human volunteers of known HLA type (Fig. 4a). High-resolution mass spectrometry (MS) analysis identified SaCas9-derived peptides presented by MHC-II proteins. The donor HLA-DR alleles and peptides identified are illustrated in Fig. 4b and a comprehensive list of all peptides identified is provided in Supplementary Data File 1 and the characteristics of the peptides associated with the MHC-II proteins are summarized in Table 1.

To determine which of the SaCas9-derived peptides identified in the MAPPs assay are biologically and potentially clinically significant, we merged the datasets for the MAPPs and T-cell proliferation assays. Figure 4c shows the peptide MHC-II combinations associated with T-cell proliferation overlaid on peptides identified in the MAPPs assay. To identify peptides which are identified in both assays we first matched the donors from the T-cell proliferation assay with donors used in the MAPPs assay. We assumed that donors were a match if they shared at least one MHC-II DRB1 allele. We compared the peptides identified in the MAPPs assay with the peptide pools found to be matching in the T-cell proliferation assay for matching donors. We combined all the matching residues across all donors and filtered out peptides shorter than nine amino acids long. Finally, we cross referenced these matching regions with distinct peptides found in the MAPPs assay. We identified 22

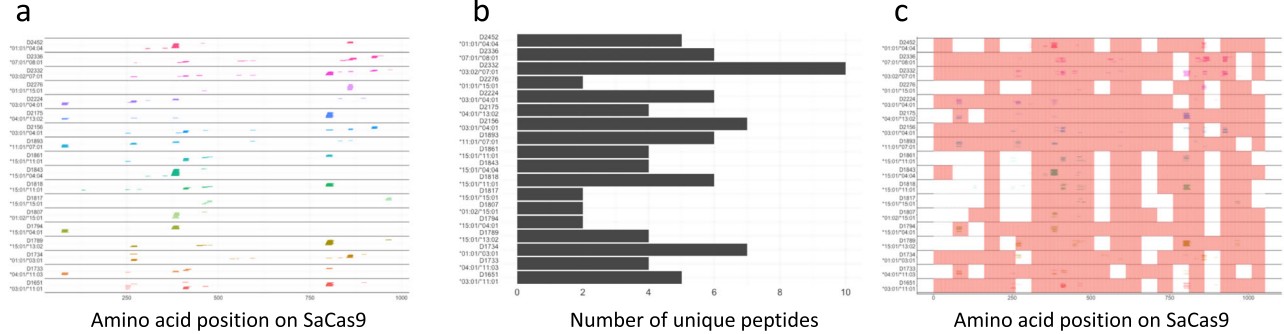

**Fig. 4 Peptides Identified in a MAPPs Assay. a** Peptides identified in a MAPPs assay are shown with their positions on the SaCas9 protein (depicted on the X-axis). The peptides identified are shown individually for each of 18 donors. The HLA-DRB1 alleles associated with the donors are depicted on the Y-axis. The peptides are stacked to show multiple peptides detected at each position on the Cas9 sequence for each donor. **b** The number of unique, continuous SaCas9 peptides detected on DCs from each donor. **c** Results of the MAPPs assay (colored lines) are overlaid on the results of the flow cytometry-based T-cell proliferation assay (pink areas). We assumed that donors were a match if they shared at least one HLA allele.

## Table 1 Summary of Cas9-derived peptides identified in the MAPPs assay.

| | |
|---|---|
| Number of donors | 18 |
| Number of unique peptides | 26 |
| Minimum peptide length | 7 |
| Maximum peptide length | 24 |
| Median peptide length | 15 |
| Number of peptides | 681 |
| Avg peptides per donor | 37.8 |

## Table 2 Cas9 peptides associated with MHC-II molecules on DCs that also elicit T-cell responses.

| Peptide sequence | Amino acid position |
|---|---|
| LFDYNLLTDHSELSGINPYEARV | 71–93 |
| SVKYAYNADLYNALNDL | 246–262 |
| NADLYNALNDLNNLVITRDENEKLE | 252–276 |
| KEILVNEEDIKGYR | 301–314 |
| LDQIAKILTIYQSSE | 348–362 |
| NLNSELTQEEIEQISNLKGYTGTHN | 370–394 |
| AINLILDELWHTNDNQIA | 399–416 |
| ILDELWHTNDNQIAIFNR | 403–420 |
| TNDNQIAIFNRLKLVPK | 410–426 |
| LVDDFILSPVVKRSFIQS | 440–457 |
| IQSIKVINAIIKKYGLPND | 455–473 |
| LPNDIIIELAREKNSKDA | 470–487 |
| EGKCLYSLEAIPLEDL | 531–546 |
| NYEVDHIIPRSVSFDNSFNN | 552–571 |
| TPFQYLSSSDSKISYE | 587–602 |
| KDDKGNTLIVNNLNGLYDKDNDKL | 793–816 |
| LLMYHHDPQTYQK | 827–839 |
| DEKNPLYKYYEETGNYLTKYS | 849–869 |
| GNYLTKYSKKDNGPV | 862–876 |
| LDNGVYKFVTVKNLDVIK | 918–935 |
| KENYYEVNSKCYEEAK | 936–951 |
| ISNQAEFIASFYNNDLIK | 956–973 |

peptides likely to be biologically and clinically important in eliciting immune responses to SaCas9 (Table 2). These 22 peptides are proteolytically processed by the DCs, presented on MHC-II proteins and activate CD4⁺ T-cells. These 22 peptides were found to be associated with MHC-II variants, which have a cumulative frequency of 69.6% in the NA population.

## Discussion

Studies published in the last few years have demonstrated T- and B-cell responses to Cas9[5–7]. These findings highlight an important aspect in the development and licensure of clinical applications of gene editing. However, immunogenicity risk assessments that can be used as part of clinical decision making require more detailed information and a better understanding of immune responses to Cas proteins. The current study provides a comprehensive evaluation of SaCas9-derived peptides that are generated by antigen-presenting cells, presented by MHC-II proteins, and elicit proliferation of CD4⁺ T-cells.

Our data show that while individual responses to SaCas9-derived peptide pools are diverse, there are some regions of the SaCas9 protein that are more promiscuous that others. Previous exposure to the Cas9 protein may explain the observed differences in donor T-cell stimulation. This is because donors with no previous exposure to SaCas9 would be unlikely to respond to the peptide pools. Following further evaluation in a clinical setting, these variations in the fraction of peptide pools that elicit T-cell responses could be used in determining the immunogenicity risk of prospective patients before treatment. However, the current association between the stimulation of T-cell activation markers and clinical immunogenicity is poorly understood[10]. For example, the first use of CRISPR/Cas9 gene editing was reported in the context of ex vivo editing of T-cells as a cancer immunotherapy following adoptive cell transfer[1]. In a small number of patients (n = 3), both the pre-existing antibody and T-cell responses to SpCas9 were demonstrated. The investigators reported no adverse events associated with gene editing and none of the patients developed humoral responses to the infused cells and the gene-edited T-cells persisted despite baseline cellular reactivity. It should be noted that the patients in this report were immunosuppressed at the time of infusion. To help address this disparity, more comprehensive algorithms for immunogenicity risk-assessment of individual patients will need to be developed as additional nonclinical and clinical data in the same individual becomes available.

Using a MAPPs assay we identified 26, SaCas9-derived peptides that were presented by MHC-II variants on DCs obtained from 18 individual donors. Approximately 85% of these (22 peptides) also induced T-cell proliferation. This is not surprising as SaCas9 is a protein of bacterial origin, i.e., almost completely foreign to human immune systems. Thus, almost all the peptides presented by antigen-presenting cells are likely to be recognized by T-cell receptors. Importantly, any individual patient would have fewer high-risk T-cell epitopes based on that individual's unique HLA repertoire. For instance, the number of SaCas9

peptides identified on DCs from individual donors ranges from 2 to 10. Our dataset also allows the identification of MHC-II alleles that may be associated with a higher risk of immunogenicity, i.e., some MHC-II alleles engage with a larger sub-set of Cas9-derived peptide pools.

Granular data that has been presented here encapsulates many of the complexities associated with immune responses to SaCas9. However, there is a lack of additional information that is required for meaningful decisions about the immunogenicity risk for an individual patient. For instance, an important limitation of this study is that we have not evaluated peptides that activate CD8$^+$ T-cells in conjunction with SaCas9 peptides associated with MHC-I proteins. We are currently developing a fit-for-purpose workflow to investigate this. MHC-I proteins engage with intracellular proteins. This requires standardization and characterization of SaCas9 expression in antigen-presenting cells. This information will be important to obtain because recent evidence in an animal model suggests a clear role for a CD8$^+$ T-cell immunity to Cas9 in preventing effective gene editing[12]. It was demonstrated in mice that while pre-existing anti-SaCas9 antibodies did not prevent gene editing, a robust CD8$^+$ T-cell response ensued. This cytotoxic T-cell response eliminated the genome-edited cells.

The current understanding of the association between the results of in vitro and ex vivo assays, and clinical outcomes are limited. There are no adverse events related to Cas-protein immunogenicity reported up to date. However, the clinical assessment of safety and efficacy of CRISPR-Cas-mediated gene therapy would benefit from the improved science-based immunogenicity risk assessments using the tools reported here. As more clinical data emerges encompassing the relevant immune response to CRISPR-Cas proteins, these tools can be used to better profile the patient population and assist with understanding the impact of immunogenicity on clinical safety and efficacy.

## Methods

**HLA-typed peripheral blood mononuclear cells (PBMCs)**. Twenty-one HLA-typed donor PBMCs were selected using an in-house algorithm called "SampPick"[17]. The preselected donors were purchased from Cellular Technology Limited (CTL) ePBMC library. The PBMCs were stored in vapor–liquid nitrogen until further use.

**Peptides and proteins**. The individual SaCas9 peptides (>90% purity), with amino acid sequences specified in Fig. S1, were purchased from GenScript. The individually lyophilized 15-mer peptides were initially dissolved in 100% Dimethyl sulfoxide (DMSO) (Fisher Scientific) at a concentration of 10 mg/mL and further diluted with 10% DMSO in CTL test media at a final concentration of 1 mg/mL. Overlapping 15-mer peptides were staggered by five amino acids. Thus, the first peptide covered amino acids 1–15 of the SaCas9 sequence, the second amino acids 6–20, the third amino acids 11–25, and so on. The last peptide is a 13-mer peptide due to the length of the SaCas9 protein. Ten individual peptides were pooled, each containing 10 consecutive 15-mer peptides (except for pool 21, which contains nine peptides). Aliquots of pooled peptides were stored at −80 °C. SaCas9 and SpCas9 (Cas9 from *Staphylococcus pyrogens*) proteins were endotoxin-free products manufactured by Editas Medicine as previously described[6]. We also obtained a laboratory-grade SpCas9 from a commercial supplier as a control. Peptides and proteins were used at 10 μg/mL concentration for all experiments.

**Cell culture**. Cryopreserved peripheral blood mononuclear cells (PBMCs), which were purchased from CTL, were thawed by following the supplier's instructions. Briefly, Cryopreserved PBMCs were thawed at 37 °C by placing in a water bath for 8 min and flipping the cryovial twice to resuspend the cells. The resuspended cells were added in 10 mL of prewarmed RMNI-1640 with 1X Antiaggregate wash medium (CTL). The cells were centrifuged at room temperature at 330 g for 10 min and the supernatant was discarded. To completely remove the freezing media, we washed the cells again with 10 mL of antiaggregate wash medium. The PBMCs were then cultured in CTL Test Medium (CTL) supplemented with L-glutamine (1%, Thermo Fisher) and costimulatory reagent (BD Bioscience) CD28/CD49d costimulatory antibody at 10 μL/mL.

**In vitro peptide stimulation**. 1.5 million PBMCs were stimulated with individual peptide pools, the full-length SaCas9 protein or Staphylococcal Enterotoxin B (SEB) (positive control, 1 μg/mL) (Sigma). The negative control (unstimulated) samples were treated with 10% DMSO in CTL Test media (DMSO working concentration 0.01%). All samples were incubated for 18 h at 37 °C, 5% CO$_2$ in a 24 well plate. For analysis of antigen-induced intracellular markers (IFN-γ, TNF-α, and IL-2) 10 μg/mL of brefeldin A (Sigma) was added 6 h postantigen stimulation.

**Intracellular cytokine staining/flow cytometry**. After stimulation, cells were collected, washed twice with 1X PBS (Thermo Fisher), and stained with fluorescently conjugated monoclonal antibodies from BD Bioscience for extracellular markers CD3 (APC-Cy7, clone SK7), CD4 (BV711, clone SK3), CD8 (BV786, clone RPA-T8) for 30 minutes at 4 °C. Antibodies were used at a dilution of 1:60. To exclude dead cells from analysis, LIVE/DEAD Fixable Aqua Dead Cell Stain (405 nm excitation) (Thermo Fisher Scientific) was added. Afterwards, cells were fixed and permeabilized using the eBiosciences FoxP3/Transcription Factor Staining Buffer Kit (Thermo Fisher Scientific) according to the manufacturer's protocol. Samples were stained for 30 min at 4 °C using fluorescently conjugated monoclonal antibodies from BD Bioscience for intracellular markers: IFN-γ (BV421, clone B27), TNF-α (PE-CF594, Mab11), and IL-2 (BV650, clone 5344.111). All antibodies were used at a dilution of 1:60. Cells were finally resuspended in 300 μL Flow Cytometry staining buffer (PBS + 1% FCS ice cold) and measured on a BD LSR Fortessa X20 flow cytometer. Compensation was performed using tubes of Ultra Comp eBeads (Thermo Fisher), individually stained with each fluorophore and compensation matrices were calculated with FACSdiva.

**Flow cytometric analysis**. Flow cytometry data collection was performed with BD FACSDiva (v6) and analysis was performed using the FlowJo software version 10.5.3. Lymphocytes were gated based on FSC verse SSC profile, and doublets were excluded using FSC-A plotted against FSC-Height (FSC-H). Once single cells were selected, the samples were further gated for alive cells, then CD3+ , CD4$^+$ T-cells for analysis of various intracellular markers (IFN-γ, TNF-α, and IL-2). Unstimulated cells were used as controls.

**In vitro profiling of peptides displayed by MHC-II after uptake by monocyte-derived dendritic cells (DCs)**. The study was performed by ProImmune (Oxford, UK) using the ProPresent Antigen Presentation Assay (see[20–24]). Monocytes were isolated from the mononuclear cell fraction of peripheral blood samples by positive selection and differentiated in vitro into immature DCs. The immature DCs were: i) Cultured in the presence of SaCas9. ii) Matured in the presence of the same protein. The mature DCs were harvested and lysed. Immune-affinity columns were used to isolate MHC-II-DR molecules from the cell lysate. The MHC-II-bound peptides were eluted and processed for further analysis by high-resolution sequencing-MS (LC-MS/MS). The resulting data were analyzed using sequence analysis software referencing the Swiss-Prot Human Proteome Database with the incorporated test item sequence(s). LC-MS/MS-based identification of peptide sequences was based on scoring algorithms and statistical significance determination. The likelihood of peptides to be real identities is described by their Expect Value (EV) and by the False Discovery Rate (FDR) obtained by searching an unrelated database. The EV considers specific features of the experimental MS/MS spectrum, e.g., the number of peaks that match predicted fragments from a peptide sequence in the database. The EV provides a statistical significance for the peptide identification and it is defined as being reflective of the number of assignments with equal (or better) scores that are expected to occur by chance alone. The lower this value, the higher is the probability the assignment is correct. EV ≤ 0.05 was used to report significant hits as recommended by current guidelines for documentation of peptide and protein identification by mass spectrometry[24]. A false-discovery rate (FDR) is determined by repeating the search using identical searching parameters against a "decoy" database, where the sequences have been reversed or randomized. The number of matches that are found with the decoy database is an estimate of the number of false positives that are present in the results from the real or "target" database. The FDR can be expressed as the total false positives divided by the sum of true positives and false positives. Identifications reported with EV ≤ 0.05 typically have FDRs < 1%.

**Statistical test to determine positive responders**. Positive responses were determined using a one-sided Fisher's Exact Test. For each donor, each gating strategy identified a cell count for each biomarker for each pool for CD4cells. In addition, for each donor, there was a cell count for the unstimulated sample for CD4$^+$ for each gating strategy. In addition, each pool including the unstimulated well had a specific count of total CD4$^+$ cells for that donor/pool (including unstimulated) combination. Using Fisher's exact test, a p value was calculated for the one-sided hypothesis that the ratio of cells counted to total cells for a particular biomarker were higher than the ratio of cells counted to total cell for the unstimulated values for that Donor/CD cell type. These p values were adjusted for multiple comparisons using the "Holm-Bonferroni" method. Responses with a false-discovery rate < 0.05 were considered positive responders for that cell-type/pool/marker.

**Data analysis**. Data Analysis was done in R v3.6.1. Base R functions were used for Fisher testing. Graphics were created using ggplot2 (v3.3.0), cowplot (v1.0.0), reshape2 (v1.4.4). Code is available from the corresponding author with a request. Flow Cytometry Data Analysis was performed with FlowJo (v10.5.3). SampPick Software to select a representative donor cohort can be found at https://www.github.com/fda/SampPick.

**Reporting Summary**. Further information on research design is available in the Nature Research Reporting Summary linked to this article.

## Data availability

The data generated in this study have been deposited in the Harvard Dataverse Repository and can be accessed at https://doi.org/10.7910/DVN/O4PUO1. The following 5 files have been provided in the repository. File 1: Background frequencies of North American HLA-DRB1 alleles; File 2: Flow cytometry, MFI for cytokines; File 3: Flow cytometry, cell counts for each marker; File 4: HLA typing of donors used as a source of cells for follow cytometry; File 5: Mass spectrometric data for the MHC Associated Peptide Proteomics assay. The datasets used to generate each of the figures are: Fig. 1a (Files 1, 4, 5); Fig. 1b,c & Fig. 2a,b (Files 2, 3); Fig. 3 (Files 1, 2, 3, 4); Fig. 4a,b (File 5); Fig. 4c (Files 2, 3, 4, 5).

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

## Acknowledgements

Z.E.S. is funded by intramural grants from the US Food and Drug Administration. L.H. and J.R.M. were supported by an appointment to the Research Participation Program at CBER, US Food and Drug Administration, administered by the Oak Ridge Institute for Science and Education through an interagency agreement between the US Department of Energy and FDA. The authors thank Rick Morgan for thoughtful comments and Daniel Lagasse for assistance with preparing and reviewing the manuscript.

## Author contributions

Z.E.S. conceptualized the study. V.L.S and L.H. performed experiments. Z.E.S., V.L.S., J.M., and L.H. designed experiments. Z.E.S., V.L.S., J.M., L.H., B.R.D., S.M. and K.Z. analyzed the data. Z.E.S., V.L.S., J.M., S.M. and L.H. wrote the manuscript. Z.E.S obtained funding.

## Competing interests

Z.E.S., V.L.S., J.M., and L.H. have no financial competing interests. K.Z., B.R.D. and S.M. are employees of Editas Medicine.
