## [Peer Review File · Nature Communications]

Reviewers' Comments:

Reviewer #2:

Remarks to the Author:

In this manuscript, the authors assess the T cell responses toward *Staphylococcus aureus* Cas9 peptides in a representative cohort for the North American population (n=21 healthy human adults). Using 21 peptide pools that cover the whole SaCas9 sequence, they try to correlate HLA-type and T cell responses toward one of the different pools, ergo distinct fragment or "hot-spot" within the SaCas9 protein.

The work is novel and important for the field of gene therapy. The selection of the donor cohort was well done. The workup of their stimulation results was thorough and in depth, which could add value to the field. While the stimulation results are interesting, there are some technical concerns that the authors should re-evaluate, especially since the overall frequency of responders to SaCas9 whole protein stimulations do neither match previous observations (Charlesworth et al 2019, Wagner et al 2019) nor their own peptide stimulation experiments. This is particularly surprising, because they included a co-stimulatory agent and could not find a single "true" responder that shows CD4+ activation, and only three donors which either upregulated IFN γ or CD154. While the peptide responses were generally higher, with an overall frequency of responders of 10/21 (CD4+ or CD8+ response to at least 1 peptide).

The most important part of the work is the attempted epitope mapping in different HLA-settings. For the broad assessments with 21 pools of each 10 peptides, more donors would be needed to clearly elucidate any HLA-haplotype specific responses. An in-depth characterization of the distinct single epitope responsible for a T cell response within a given 10 peptide pool would be an important addition to strengthen this manuscript. Furthermore, the title is misleading. It should be limited to the *Staphylococcus aureus* Cas9. Additionally, no adequate controls were performed to fully support the claim that their findings were MHC-restricted (this is of course standard knowledge, but should be evaluated in a few donors by addition of MHC-blocking antibodies or use of MHC-KO APC).

Importantly, all presented data rely on ex vivo stimulations and no restimulation assays or functional assessments (killing of gene edited target) were performed. Such assays would greatly benefit and support the overall inherent validity of the study.

Technical concerns:

The methods do not state how the PBMCs were cryopreserved and how much time the cells were rested prior to activation. DMSO in freezing medium can inhibit T cell activation (<https://pubmed.ncbi.nlm.nih.gov/30201392/>). It is therefore important to disclose the time when the cells were stimulated (after freezing and thawing).

The authors are trying to characterize rare events, therefore 1.5 Million PBMCs may cut it very close. Generally, responses to Cas9 protein are in the magnitude order of 1% to 0,01% of the respective T cells subset (CD4/CD8), subdividing this into 20 different pools may easily reach the limit of detection. Notably, the authors do not state how many events were collected which seems important to me when fishing for rare T cell clones.

Regarding the flow cytometry panel, I believe their low CD137 response could be related to the use of the BV605 fluorochrome. Generally, CD137 does not allow clear distinguishing of CD137 high and low expression, which is important as the previous report by Wagner et al actually showed that CD137 low/intermediates contain most of the Treg (somewhere in supplements).

What is the reason for adding BFA after 6 hours in peptide stimulation experiments? Peptide proteolysis should be much faster, potentially you miss some of the response or there could be T cell fratricide happening in case of potent CD8+ T cell responses.

Detailed comments:

Title) very broad, the authors only analyze SaCas9, so please refrain from generalization. Subtitle is irrelevant and misleading, because they did not show a single functional assay where they demonstrate that the cells they identify can actually kill / expand or can be re-stimulated by gene edited target cells.

Figures:

1a) fine

1b) should go into supplementary figure, also alive/dead gate example seems to cut in the dead Aqua positive cell population...

1c +d) use different format (e.g. large dots) as most readers will not be able to discern the few small dots; also please add in frequencies in gate in a bigger font. Gating also looks questionable, why does the base population shift toward the right in the IFN γ axis? Compensation problems? Revisit, also provide some pictures IFN γ vs TNF α or similar for supplementary figures. (Best responder donor 9 e.g.?)

1e+f) I like it, here I can also distinguish between the different colors. Some people with green-red blindness may not?

1g+h) very low overall frequency, but figure is fine

2a+b) cannot distinguish between the colors 1+2, 21 and SaCas9. This might be me though

2c+d) very good and interesting way to show this. The underlying data should be made available.

3) overall: This is a good idea to do but would likely benefit from more donors to see whether trends are similar in more than just 10 responders.

Text:

Line 24: authors should add in Stadtmauer et al (Science 2020) as well. Further, add "in humans" and "preexisting" to this sentence.

Line 28: authors should mention Ferdosi et al, as they did this for *S. pyogenes* Cas9 with in silico methods. Also review by Chew (2018; <https://doi.org/10.1002/wsbm.1408>) features in silico analysis for potential MHC binding of SaCas9 epitopes

Line 29: MHC-dependent: very true, but whether SaCas9 in some of your responding donors is MHC dependent (e.g. add MHC blocking antibodies during stimulation)

Line: 74+75: "Previous exposure to the Cas9 protein may explain differences in donor T cell stimulation, particularly for those that did not respond to the peptide pools." this sentence needs to be revisited. Why should previous exposure to Cas9 explain that some of your donor did not respond to the peptide pools.

Line 82-86: here the authors dive into speculation, maybe reserve some for the discussion
Paragraph starting in line 87: very important, but please show MHC dependent activation in your setting.

Line 104: "Thee MHC-I classes" should be "three MHC-I classes"

Line 109-113: good point. But why didn't you observe responses to Cas9 protein in your experiments?

Line 128: wouldn't this be expected with such a large protein?

Line 130-131: the authors did not identify a single epitope, but rather fragments within the Cas9 proteins that contain an epitope. Complete epitope mapping would entail mapping which of the respective 10 peptides actually contain the immunogenic epitope.

Line 133-135: true, but the data presented is not sufficient to actually perform immunosilencing, because the exact "immunodominant" epitopes within a single individual remains unknown with the assays provided here

Line 137-138: starting from "moreover," but are there variants that have a higher risk? More donors are needed to validate this claim.

Line 139: comment on no/little response during SaCas9 protein stimulation needed

Line 152-154: A bit simplistic, Crosspresentation occurs, also CD4 T cell with cytotoxic function have been described in gene therapy

Line 159-163: MAPPs would be great to include, but is probably out of scope for this study.

Line 169-170: Stadtmauer et al found T cell responses in 2 of 3 patients.

Important addition to the discussion would be the recent preclinical in vivo gene model with preexisting immunity to SaCas9. Li and colleagues showed that SaCas9 protein immunization prior to AAV-Cas9 targeting of the liver reduced transgene expression in the liver and no edited cells were left 12 weeks after AAV infusion. This is one of the strongest preclinical evidences that we need to evaluate preexisting immunity in humans...

Line 175: How does this study help to develop other delivery technologies? Rephrase or delete

Line 181-184: Very long sentence, theoretically true, but data presented by authors is not sufficient to predict risk through HLA-haplotype.

Line 202: see technical concern, 1.5 million PBMCs

Line 208: Flow cytometry – how many cells were analysed?

Line 216: CD137 marker – fluorochrome not very bright; better alternative might be PE or conjugates or APC/AF647...

Reviewer #3:

Remarks to the Author:

This manuscript asks an important question in the field of gene therapy: what is the pre-existing immunity to the *S. Aureus* Cas9 protein? The hypothesis is that pre-existing T cell immunity may impact toxicity associated with Crispr-based gene therapies.

Strengths of the study: A systematic approach to measuring T cell immunity by testing pools of overlapping peptides, multi parametric flow cytometry for both CD4 and CD8 immunity, and a well characterized and HLA-representative set of PBMCs used for this study (excellent). Positive controls are present to demonstrate immune stimulation

Weaknesses: By relying on peptide pools, the individual epitope contributions may be diluted, and follow up analysis of the individual epitopes (and prediction of the specific epitopes) is not performed. As these do not involve in vitro stimulation, it may be under representing low frequency responses.

As the authors note, there are several prior manuscripts in this field (Charlesworth et al, Wagner et al, Ferdosi et al) that identify pre-existing T cell immunity to Cas proteins in healthy donors. This manuscript does note several pools of peptides which induce CD4 or CD8 immunity, but further characterization is not performed, and the overall immune responses appear weaker than previously reported. The manuscript would be strengthened if the individual epitopes and HLA restriction were identified, and comparison to previous epitopes were discussed.

Overview of major changes to the manuscript for all reviewers:

A major concern was that our study did not identify specific epitopes but rather regions of the SaCas9 protein which contained epitopes. We share this concern and have completely overhauled this study. The study now endeavors to systematically identify biologically and clinically relevant peptide epitopes. Part 1 uses the data presented in the earlier iteration of the manuscript generated by intracellular cytokine levels in T-cells following stimulation with individual peptide pools. For Part 2, we identified SaCas9 derived peptides on MHC-II proteins on dendritic cells (DCs) following incubation with the full-length SaCas9. This is a novel data set for any Cas9 protein. No such data is available in the public domain. In Part 3 we combined the two data sets to estimate a set of SaCas9 derived peptides that are: (i) Generated by proteolysis of SaCas9 in DCs, (ii) presented by MHC-II proteins on DCs and can elicit T-cell proliferation. This strategy we believe is most likely to allow identification of peptides of biological and clinical importance.

The strategy we have presented here is currently useful in identifying peptides that engage with MHC-II cells and stimulate CD4⁺ T-cells. This is because to identify peptides presented by MHC-I proteins the SaCas9 protein needs to be expressed intracellularly in antigen presenting cells. The standardization and characterization of the expression system and the MAPPs assay to obtain consistent and reliable results for peptides presented by MHC-I molecules will take some time and is thus outside the scope of this submission.

Reviewer #2:

In this manuscript, the authors assess the T cell responses toward Staphylococcus aureus Cas9 peptides in a representative cohort for the North American population (n=21 healthy human adults). Using 21 peptide pools that cover the whole SaCas9 sequence, they try to correlate HLA-type and T cell responses toward one of the different pools, ergo distinct fragment or “hot-spot” within the SaCas9 protein. The work is novel and important for the field of gene therapy. The selection of the donor cohort was well done. The workup of their stimulation results was thorough and in depth, which could add value to the field. While the stimulation results are interesting, there are some technical concerns that the authors should re-evaluate, especially since the overall frequency of responders to SaCas9 whole protein stimulations do neither match previous observations (Charlesworth et al 2019, Wagner et al 2019) nor their own peptide stimulation experiments. This is particular surprising, because they included a co-stimulatory agent and could not find a single “true” responder that shows CD4⁺ activation, and only three donors which either upregulated IFN γ or CD154. While the peptide responses were generally higher, with an overall frequency of responders of 10/21 (CD4⁺ or CD8⁺ response to at least 1 peptide).

RESPONSE: We thank the reviewer for the comments about the potential importance of this study. There are two reasons why the overall frequency of responders is lower than reported previously: (i) The criterion we use to identify a donor as positive. (ii) The statistical methods we used. Charlesworth et al. for instance established “a worst-case scenario”; a donor was deemed

a responder if cells from that donor tested positive for even one of the 3 cytokines measured. We on the other hand used a rule wherein 3 of the 5 markers we measured had to test positive for a donor to be deemed positive. We also used what we believe is a more robust statistical method for determining a positive signal for a donor-protein/peptide combination. The cell counts for each Donor/Protein/cytokine was compared to its respective unstimulated value. Importantly, the p values were **adjusted** using the Bonferroni method. We have used adjusted p-values in our analysis to avoid spurious false-positive rates due to multiple testing. The experimental setup consists of 21 donors, 3 proteins and 5 biomarkers, we would expect that roughly 16 ($315 \cdot .05$) of these results would be significant based on chance alone. The adjusted p-values helps to control the for false positives that were introduced due to the multiple analyses being run. Below we compare results for three Cas9 proteins (Sa- and Sp-Cas9 proteins manufactured under GMP conditions by Editas) and a commercially available (“off the shelf”) Sp-Cas9. There is a decrease in the frequency of responders (b) when we use adjusted P-values.

We have consistently found that the Cas9 proteins from Editas show a lower frequency of responders compared to commercially available preparations. Using unadjusted P-values and the criterion for identifying a responder (a positive signal for any cytokine in the ICS experiment), the frequency of responders for a commercially available Cas9 protein was ~75% which is actually higher than that observed in the papers by Chatsworth et al (44%).

Given the current state-of-the art where reagents are not well characterized and assays are not validated, it is inevitable that there will be discrepancies with respect to overall frequencies of positive responders.

In the revised manuscript we have reported the results for T cell activation based on the three cytokine markers but changed our criterion for identifying a responder as responsive to at least one marker. We do this acknowledging that we can expect some over-identification of responding peptides. However, we will continue to utilize adjusted P-values as this is the appropriate statistical methodology. We gave also included data (not included in the previous version of the manuscript) for a SpCas9 manufactured by Editas and a SpCas9 obtained from a commercial manufacturer of laboratory grade Cas9 proteins.

The discrepancy between the results obtained with individual peptides and SaCas9 protein represents a limitation of the assay (which has been addressed in the revised submission). Biologically relevant epitopes are peptides that have been processed, presented on MHC proteins, and stimulate T-cells. The use of “pre-formed” peptides fails to provide information about whether the proteolytic machinery of antigen presenting cells can generate these peptides. To address this issue, we have carried out an MHC Associated Peptide Proteomics (MAPPs) assay wherein following incubation of dendritic cells with SaCas9, MHC proteins are isolated, and peptides derived from Cas9 identified by mass spectrometry. The revised manuscript combines the results of the MAPPs and T-cell proliferation assays to identify biologically and clinically relevant epitopes.

The most important part of the work is the attempted epitope mapping in different HLA-settings. For the broad assessments with 21 pools of each 10 peptides, more donors would be needed to clearly elucidate any HLA-haplotype specific responses. An in-depth characterization of the distinct single epitope responsible for a T cell response within a given 10 peptide pool would be

an important addition to strengthen this manuscript. Furthermore, the title is misleading. It should be limited to the *Staphylococcus aureus* Cas9. Additionally, no adequate controls were performed to fully support the claim that their findings were MHC-restricted (this is of course standard knowledge but should be evaluate in a few donors by addition of MHC-blocking antibodies or use of MHC-KO APC). Importantly, all presented data rely on ex vivo stimulations and no re-stimulation assays or functional assessments (killing of gene edited target) were performed. Such assays would greatly benefit and support the overall inherent validity of the study.

RESPONSE: To address the concerns about MHC-restriction and identification of individual epitopes from the peptide pools we have taken a slightly different track than that suggested by the reviewer. As most peptide pools elicit responses; an experiment where T-cell proliferation assays are carried out using 209 peptides and 20 (or greater as suggested by the reviewer) donors would be logistically challenging. More importantly the information would not be very useful. As stated in the response to Comment 1, just because a peptide stimulates a T cell response does not mean it is biologically relevant. Proteolysis of the protein, loading and presentation on MHC and stimulation of T cells are all necessary to elicit an immune response. Thus, we have carried out a MAPPs assay (to determine processing of peptide) in addition to the T-cell proliferation assay (to determine if the presented peptide elicits an immune response). This has unfortunately limited the scope of the manuscript to peptides that stimulate CD4⁺ T-cells. The development of a MHC-I MAPPs assay that is reliable and gives consistent results is a complex endeavor and these studies have been initiated.

Technical concerns:

The methods do not state how the PBMCs were cryopreserved and how much time the cells were rested prior to activation. DMSO in freezing medium can inhibit T cell activation (<https://pubmed.ncbi.nlm.nih.gov/30201392/>). It is therefore important to disclose the time when the cells were stimulated (after freezing and thawing).

RESPONSE: Thank you for raising this important technical issue; the revised manuscript provides the detailed information sought. To answer the reviewer's questions, we did not follow the resting protocol. However, our thawing protocol contains 10 volumes of anti-aggregate wash media and double the washing steps should clear the DMSO from the cells. In our stimulation assays, we have used a maximum 0.01% DMSO. Based on the publication you refer to, this concentration of DMSO has no impact on T-cell assays.

The authors are trying to characterize rare events, therefore 1.5 Million PBMCs may cut it very close. General, responses to Cas9 protein are in the magnitude order of 1% to 0,01% of the respective T cells subset (CD4/CD8), Subdividing this into 20 different pools may easily reach the limit of detection. Notably, the authors do not state how many events were collected which seems important to me when fishing for rare T cell clones.

RESPONSE: We agree that we are characterizing rare events. With respect to the number of events, 1.5 Million PBMCs from the donors were not subdivided into the 20 conditions. 1.5 Million PBMCs were used for each experimental condition for stimulation, and 100,000 events were collected per condition during flow cytometry.

Regarding the flow cytometry panel, I believe their low CD137 response could be related to the use of the BV605 fluorochrome. Generally, CD137 does not allow clear distinguishing of CD137 high and low expression, which is important as the previous report by Wagner et al actually showed that CD137 low/intermediates contain most of the Treg (somewhere in supplements).

RESPONSE: The revised paper only reports measurements of the 3 cytokines IFN- γ , TNF- α , and IL-2. As we are following up with a MAPPs assay, we are using the intracellular cytokine staining to report a worst-case scenario with likely false positives. In the follow up computation, many of the peptides identified will be eliminated based on their absence in the results from the MAPPs assay.

What is the reason for adding BFA after 6 hours in peptide stimulation experiments? Peptide proteolysis should be much faster, potentially you miss some of the response or there could be T cell fratricide happening in case of potent CD8+ T cell responses.

RESPONSE: Wagner et al. added BFA after 6 hours in their protein stimulation experiments. For consistency and to compare our own peptide and protein stimulation results, we decide to follow their established approach.

Detailed comments:

Title) very broad, the authors only analyze SaCas9, so please refrain from generalization. Subtitle is irrelevant and misleading, because they did not show a single functional assay where they demonstrate that the cells they identify can actually kill / expand or can be re-stimulated by gene edited target cells.

RESPONSE: Per the reviewer's comment and the significant changes in the experimental strategy we have modified the title to, "*Identification of Staphylococcus aureus Cas9 derived peptides presented by MHC Class II that elicit proliferation of CD4+ T-cells*" which we believe accurately reflects this study.

Figures:

1a) fine

1b) should go into supplementary figure, also alive/dead gate example seems to cut in the dead Aqua positive cell population...

RESPONSE: As suggested by the reviewer we have moved this figure to the Supplementary information section. The gate for live dead was established by our unstimulated control. Each donor was individually reviewed to ensure that the aqua positive (dead) cells did not contribute to the analyzed population.

1c +d) use different format (e.g. large dots) as most readers will not be able to discern the few small dots; also please add in frequencies in gate in a bigger font. Gating also looks questionable, why does the base population shift toward the right in the IFN γ axis? Compensation problems? Revisit, also provide some pictures IFN γ vs TNF α or similar for supplementary figures. (Best responder donor 9 e.g.?)

RESPONSE: The figure has been modified per the reviewer's suggestion. The base population was shifted for visual purposes. The base population sometimes sits along the y-axis with

certain markers. To prevent crowding, the scale was slightly adjusted. A similar scaling is shown in Figure 1a of Wagner et al.

1e+f) I like it, here I can also distinguish between the different colors. Some people with green-red blindness may not?

RESPONSE: As suggested by the reviewer we have changed colors to make the figure suitable for individuals with green-red blindness. Revised manuscript only has data for stimulation of CD4⁺ cells and the 3 cytokines.

1g+h) very low overall frequency, but figure is fine

RESPONSE: The overall frequency has been addressed in the Response to Comment 1. Using the revised methodology, we show a higher overall frequency for peptides as well as some Cas9 proteins.

2a+b) cannot distinguish between the colors 1+2, 21 and SaCas9. This might be me though

RESPONSE: We have changed the colors and each pool can be distinctly seen.

2c+d) very good and interesting way to show this. The underlying data should be made available.

RESPONSE: The underlying data has been provided as a file in the supplementary section.

3) overall: This is a good idea to do but would likely benefit from more donors to see whether trends are similar in more than just 10 responders.

RESPONSE: While the promiscuity score would no doubt be more accurate with a larger data set, we currently have a good representation of MHC-II DRB 1 variants in the experimental cohort.

Text:

Line 24: authors should add in Stadtmauer et al (Science 2020) as well. Further, add “in humans” and “preexisting” to this sentence.

RESPONSE: We have added the reference and altered the sentence.

Line 28: authors should mention Ferdosi et al, as they did this for *S. pyogenes* Cas9 with in silico methods. Also review by Chew (2018; <https://doi.org/10.1002/wsbm.1408>) features in silico analysis for potential MHC binding of SaCas9 epitopes

RESPONSE: We have made the changes suggested by the reviewer.

Line 29: MHC-dependent: very true, but whether SaCas9 in some of your responding donors is MHC dependent (e.g. add MHC blocking antibodies during stimulation)

RESPONSE: The new data with the MAPPs makes this experiment redundant.

Line: 74+75: “Previous exposure to the Cas9 protein may explain differences in donor T cell stimulation, particularly for those that did not respond to the peptide pools.” this sentence needs to be revisited. Why should previous exposure to Cas9 explain that some of your donor did not respond to the peptide pools.

RESPONSE: We have altered the sentence, which now reads, “Previous exposure to the Cas9 protein may explain differences in donor T cell stimulation, donors not having previous exposure to Cas9 would be unlikely to respond to the peptide pools.”

Line 82-86: here the authors dive into speculation, maybe reserve some for the discussion
Paragraph starting in line 87: very important, but please show MHC dependent activation in your setting.

RESPONSE: As suggested we have deleted the speculative statement. We also believe that by directly identifying peptides presented by MHC proteins we have strong evidence of MHC engagement with Cas9 peptides.

Line 104: “Thee MHC-I classes” should be “three MHC-I classes”

RESPONSE: This statement was deleted during revision.

Line 109-113: good point. But why didn’t you observe responses to Cas9 protein in your experiments?

RESPONSE: As the newly incorporated MAPPs data shows (Figure 4c, pink area) many peptides stimulate T-cells however these peptides are not presented by MHC molecules when the full-length SaCas9 is incubated with dendritic cells.

Line 128: wouldn’t this be expected with such a large protein?

RESPONSE: We apologize but we don’t understand the question.

Line 130-131: the authors did not identify a single epitope, but rather fragments within the Cas9 proteins that contain an epitope. Complete epitope mapping would entail mapping which of the respective 10 peptides actually contain the immunogenic epitope.

RESPONSE: In the revised manuscript we identify 26 peptides that are biologically important epitopes for MHC-Class II mediated stimulation of CD4⁺ T cells.

Line 133-135: true, but the data presented is not sufficient to actually perform immunosilencing, because the exact “immunodominant” epitopes within a single individual remains unknown with the assays provided here Line 137-138: starting from “moreover,” but are there variants that have a higher risk? More donors are needed to validate this claim.

RESPONSE: We concur and have removed these statements from the revised manuscript.

Line 139: comment on no/little response during SaCas9 protein stimulation needed

RESPONSE: We have provided you with analyses that provides an explanation for the differences in response to the Cas9 protein. Until there is a consensus on quality control of all reagents (including Cas9 proteins), protocols and methods, statistics for identifying donor cells as eliciting a positive T cell response there will continue to be discrepancies.

Line 152-154: A bit simplistic, Cross presentation occurs, also CD4 T cell with cytotoxic function have been described in gene therapy.

RESPONSE: We agree and have removed this discussion.

Line 159-163: MAPPs would be great to include but is probably out of scope for this study.

RESPONSE: We have carried out and included MAPPs data for MHC Class II presentation. The trade off (at least at this point) is that this strategy works only for identifying Cas9 peptides associated with MHC-II mediated stimulation of CD4⁺ T-cells.

Line 169-170: Stadtmauer et al found T cell responses in 2 of 3 patients.

Important addition to the discussion would be the recent preclinical in vivo gene model with preexisting immunity to SaCas9. Li and colleagues showed that SaCas9 protein immunization prior to AAV-Cas9 targeting of the liver reduced transgene expression in the liver and no edited cells were left 12 weeks after AAV infusion. This is one of the strongest preclinical evidences that we need to evaluate preexisting immunity in humans...

RESPONSE: Thank you very much for bringing this important study to our attention, we have discussed this study in the revised manuscript.

Line 175: How does this study help to develop other delivery technologies? Rephrase or delete

RESPONSE: We have deleted this statement.

Line 181-184: Very long sentence, theoretically true, but data presented by authors is not sufficient to predict risk through HLA-haplotype.

RESPONSE: We have deleted this statement.

Line 202: see technical concern, 1.5 million PBMCs

RESPONSE: Please see response to technical concern, above.

Line 208: Flow cytometry – how many cells were analysed?

RESPONSE: 100,000 events were collected per condition during flow cytometry.

Line 216: CD137 marker – fluorochrome not very bright; better alternative might be PE or conjugates or APC/AF647...

RESPONSE: We thank the reviewer for this useful information and will incorporate it in our experimental designs moving forward.

Reviewer #3:

This manuscript asks an important question in the field of gene therapy: what is the pre-existing immunity to the S. Aureus Cas9 protein? The hypothesis is that pre-existing T cell immunity may impact toxicity associated with Crispr-based gene therapies.

Strengths of the study: A systematic approach to measuring T cell immunity by testing pools of overlapping peptides, multi parametric flow cytometry for both CD4 and CD8 immunity, and a well characterized and HLA-representative set of PBMCs used for this study (excellent). Positive controls are present to demonstrate immune stimulation

Weaknesses: By relying on peptide pools, the individual epitope contributions may be diluted, and follow up analysis of the individual epitopes (and prediction of the specific epitopes) is not performed. As these do not involve in vitro stimulation, it may be underrepresenting low frequency responses.

RESPONSE: We have addressed the concern regarding identification of specific epitopes expressed by both reviewers. Please see our response to Comment 2 from Reviewer 1 for details. We carried out an MHC Associated Peptide Proteomics (MAPPs) assay wherein following incubation of dendritic cells with SaCas9, MHC proteins are isolated, and peptides derived from Cas9 identified by mass spectrometry. The revised manuscript combines the results of the MAPPs and T-cell proliferation assays to identify biologically and clinically relevant epitopes.

As the authors note, there are several prior manuscripts in this field (Charlesworth et al, Wagner et al, Ferdosi et al) that identify pre-existing T cell immunity to Cas proteins in healthy donors. This manuscript does note several pools of peptides which induce CD4 or CD8 immunity, but further characterization is not performed, and the overall immune responses appear weaker than previously reported. The manuscript would be strengthened if the individual epitopes and HLA restriction were identified, and comparison to previous epitopes were discussed.

RESPONSE: We have included the MAPPs assay to identify biologically relevant epitopes.

Reviewers' Comments:

Reviewer #2:

Remarks to the Author:

The manuscript currently available is significantly better. I must confess that it is almost a completely new manuscript and I very much appreciate how deeply and critically the authors have revised their work. For this reason, I feel that the authors have, for the most part, answered my comments satisfactorily. I would suggest that the authors continue to refine and edit a few more points:

- The authors should highlight what the relevance of their data is to the field. In this sense, what relevance might an intracellular protein have for a CD4 immune response? Please discuss and maybe add to the abstract.
- I still think that the figures should be revised. The current presentation is partly not intuitive and also does not meet current standards (font sizes, color selection, resolution, style).
- One last and perhaps important point is. Unfortunately, from the written part, I do not understand how the peptides were organized. What is the coverage on the protein? Surely, the group has worked with individual peptides? SaCas9 is huge, how can the linearized peptides give coverage, especially stochastically? From the Supplementary Figure 1, it mostly becomes clear. However, you should make this clearer in text form.

Reviewer #3:

Remarks to the Author:

This is a comprehensive analysis of the specificity and frequency of CD4+ T cell immunity to SaCas9, using pools of overlapping peptides. Overall the revised manuscript is improved by detailed in greater specificity the peptides that each donor reacts to, and includes additional controls and demonstration of antigen processing.

Point-by-point responses to the reviewers' comments

Reviewer #2 (Remarks to the Author):

The manuscript currently available is significantly better. I must confess that it is almost a completely new manuscript and I very much appreciate how deeply and critically the authors have revised their work. For this reason, I feel that the authors have, for the most part, answered my comments satisfactorily. I would suggest that the authors continue to refine and edit a few more points:

Comment 1: The authors should highlight what the relevance of their data is to the field. In this sense, what relevance might an intracellular protein have for a CD4 immune response? Please discuss and maybe add to the abstract.

Response to comment 1: We have included in the discussion a paragraph on the limitations of the current study and the measurement of CD4+ T cell responses. Please see paragraph on **Page 5**, which states: *“Granular data that has been presented here encapsulates many of the complexities associated with immune responses to SaCas9. However, there is a lack of additional information that is required for meaningful decisions about the immunogenicity risk for an individual patient. For instance, an important limitation of this study is that we have not evaluated peptides that activate CD8+ T-cells in conjunction with SaCas9 peptides associated with MHC-I proteins. We are currently developing a fit-for-purpose workflow to investigate this. MHC-I proteins engage with intracellular proteins. This requires standardization and characterization of SaCas9 expression in antigen presenting cells. This information will be important to obtain because recent evidence in an animal model suggests a clear role for a CD8+ T-cell immunity to Cas9 in preventing effective gene editing 12. It was demonstrated in mice that while pre-existing anti-SaCas9 antibodies did not prevent gene editing, a robust CD8+ T-cell response ensued. This cytotoxic T-cell response eliminated the genome-edited cells.”*

Comment 2: I still think that the figures should be revised. The current presentation is partly not intuitive and also does not meet current standards (font sizes, color selection, resolution, style).

Response to comment 2: The revised figures have been organized as per the instructions to authors and they meet the resolution requirements. If the editor feels the figures need additional modifications, we will be happy to do so.

Comment 3: One last and perhaps important point is. Unfortunately, from the written part, I do not understand how the peptides were organized. What is the coverage on the protein? Surely, the group has worked with individual peptides? SaCas9 is huge, how can the linearized peptides give coverage, especially stochastically? From the Supplementary Figure 1, it mostly becomes clear. However, you should make this clearer in text form.

Response to comment 3: We apologize for the lack of clarity. In the revised manuscript we have inserted the following statement in the sub-section **Peptides and Proteins** under the section METHODS: *Overlapping 15-mer peptides were staggered by 5 amino acids. Thus, the first peptide covered amino acids 1 to 15 of the SaCas9 sequence, the second amino acids 6 to 20, the third amino acids 11 to 25 and so on. The last peptide is a 13-mer peptide due to the length of the SaCas9 protein.*

Reviewer #3 (Remarks to the Author):

Comment 1: This is a comprehensive analysis of the specificity and frequency of CD4+ T cell immunity to SaCas9, using pools of overlapping peptides. Overall the revised manuscript is improved by detailed in greater specificity the peptides that each donor reacts to, and includes additional controls and demonstration of antigen processing.

Response to comment 1: We thank the reviewer for the comment. No changes were requested.